# Test and Mesoscopic Analysis of Chloride Ion Diffusion of High-Performance-Concrete with Fly Ash and Silica Fume

**Huang Tang [1], Yiming Yang [1,*], Jianxin Peng [2,*], Peng Liu [3,*] and Jianren Zhang [2]**

[1] Hunan Engineering Research Center of Structural Safety and Disaster Prevention for Urban Underground Infrastructure, School of Civil Engineering, Hunan City University, Yiyang 413000, China; tanghuang@hncu.edu.cn

[2] School of Civil Engineering, Changsha University of Science and Technology, Changsha 410114, China; jianrenz@hotmail.com

[3] School of Civil Engineering, Central South University, Changsha 410114, China

* Correspondence: yangyiming@hncu.edu.cn (Y.Y.); jianxinpeng@csust.edu.cn (J.P.); 2015038@csu.edu.cn (P.L.); Tel.: +86-0737-4628297 (Y.Y.); +86-0731-85258698 (J.P.)

**Abstract:** High performance concrete (HPC) is a kind of concrete with mineral admixtures, which has better resistance ability to chloride ions diffusion than ordinary concrete. In the present study, the authors carried out a chloride ion diffusion experiment for the HPC with fly ash and silica fume. The influence of the water–binder ratio (W/B), binary (Portland cement–Fly ash (PC-FS) and Portland cement–Silica fume (PC-SF) and ternary (Portland cement–Fly ash–Silica fume (PC-FA-SF)) combinations on the concrete compressive strength and chloride ion diffusion was investigated. It was found that the compressive strength of normal concrete and HPC increase with the decrease in the W/B, the ratio of W/B deceasing value to strength increasing value for normal concrete is 0.74, and for the HPC is 0.20, so the influence of the W/B on the concrete strength for normal concrete was obviously more than the HPC. The influence of the contents of the SF or FA on developing the concrete strength was limited. The concrete compressive strength of ternary combination specimens decreases with the increase in FA or SF when the content of the other mineral admixture SF or FA remained unchanged. The ternary combination was more efficient in prohibiting chlorides ingress insider the specimens than the binary combination. The mesoscopic simulation and the tested value of the chloride ion under the same depth was close, the average ratio of simulation value to tested value was 0.91. The aggregate shape and distribution also had a negligible influence on chloride diffusivity in the HPC, but the chloride ion concentration increased with the increase in aggregate size.

**Keywords:** high performance concrete (HPC); chloride ion diffusion; mesoscopic simulation; fly ash; silica fume

## 1. Introduction

Many scholars have studied how adding other materials such as carbon nanotubes [1], basalt fiber [2] and mineral admixtures into normal concrete can improve the mechanical performance and durability. High performance concrete (HPC) is a kind of concrete with mineral admixtures, has high amounts of cementitious material and low water–cement ratio [3]. The addition of an appropriate amount of mineral admixture in the HPC can refine and improve the pore size distribution of the hardened cement paste, reduce the number of large pores, and increase the number of small and connected ones, which can improve the density of concrete and ultimately achieve the purpose of improving the performance of the concrete [4], so the HPC has been widely used in the erosion environment for its stronger resistance to all kinds of deleterious ions than ordinary concrete, in these ions, the aggressive of chloride ion concentration are the most obvious [5], so it is meaningful to explore the chloride ion diffusion mechanism in the HPC.

Many studies have confirmed that mineral admixtures in the HPC can improve its ability to resist chloride ion penetration compared with normal concrete, and there are two main reasons for the resistance of HPC to chloride ion penetration: firstly, the permeability of concrete mixtures, which is decreased because of the existence of the mineral admixtures. Secondly, the chloride ion binding capacity of the cementitious paste, which increases with calcium silicate hydrate gel formation and increased alumina levels inducing more chloride ion to be fixed as Friedel's salt [6]. As a common admixture, many authors focused on the durability of HPC with fly ash (FA). Sengul and Tasdemir [7] investigated the 50% replaced FA and 50% replaced finely ground granulated blast furnace slag on rapid chloride permeability. Chalee and Jaturapitakkul [8] used the original and classified FA used as a partial replacement of Portland cement at different weights of binder to investigate the effect of W/B and FA fineness on chloride diffusion coefficient ($D_c$) of concrete under a marine environment. The author concluded that the inclusion of pozzolans is more effective than decreasing the W/B to reduce the chloride permeability of concrete. Moffatt et al. [9] presented the durability performance of concrete incorporating high volumes of FA exposed to a harsh marine environment for 19 to 24 years. The depth of chloride penetration was found to be in excess of 100 mm for the concrete specimens without FA, whereas the presence of FA significantly decreased the depth of penetration to approximately 30 and 40 mm in specimens containing either normal density or lightweight aggregate, respectively. The above studies mainly focused on the influence of simply FA content on the durability performance of HPC, and a few studies deeply investigated the chloride permeability of HPC if other mineral admixtures as partial replacements of Portland cement.

At the same time, other authors explored the chloride ion penetration behavior in the HPC with other mineral admixtures. Cabrera and Nwaubani [10] measured the effective chloride ion diffusion coefficients of Portland cement (PC), PC-15% pulverized fuel ash (PFA), PC-15% metakaolin (MK) pastes using a chloride ion diffusion cell. Both the PC–MK and the PC–PFA pastes gave lower chloride ion diffusion coefficients than the PC paste, and the former gave particularly low values. Thomas et al. [11] and Hooton et al. [12] determined the chloride ion diffusion resistance of concrete replacement of PC with high-reactivity MK (HRM). The apparent diffusion coefficients were reduced on increasing the exposure time and on decreasing the W/B and showed marked decreases with increasing HRM content. Coleman and Page [13] also determined that the MK produces a significant increase in chloride ion binding capacity of the paste when used as a partial cement replacement. Seleem et al. [6] investigated the seawater resistance of concrete incorporating silica fume (SF), ground granulated blast furnace slag (GGBS), and metakaolin (MK) as an addition to cement in binary and ternary combinations. The authors confirmed that all kinds of pozzolanic materials are efficient in reducing the permeability of concrete far below the control one. Meanwhile, silica fume is the most efficient in this regard. The ternary blend (PC-MK-SF) is superior to all other mixtures in producing impermeable concrete. Geetha et al. [14] found that copper slag was marine application which could improve compressive strength and flexural strength and reduce sorptivity and chloride ion penetrability. Harilal et al's [15] work reported a novel high performance green concrete which has an better resistance ability of chloride ion than normal concrete. Jiang et al. [16] analyzed the influence of crack self-healing degree on chloride ion transport process, finding that chloride ions not only transport vertically along the crack, but also propagate in the horizontal direction of the crack. Bachtiar et al. [1] provided information about the mechanism and the influence of prepared high-performance concrete mixed with either freshwater or seawater and made to cure in freshwater, seawater, or air. Ding et al. [17] compared the different effects of FA and slag on anti-rebar corrosion ability of high strength concrete with chloride ion. FA and SF are the most common mineral admixtures used in the HPC, few above studies focused on the chloride ion diffusion behavior for HPC contained FA and SF. Though Khan [18] investigated the chloride ion penetration and diffusion for high performance concrete pulverized fuel ash and silica fume with various W/B, environmental factors such as temperature, humidity were not considered, and the

influence of cementitious material combination (FA and SF) on the chloride ion diffusion needs to be further explored.

The corrosion of steel bar induced by chloride erosion is the main factor leading to the degeneration of structure mechanical performance. In recent years, many studies have focused on the mechanical property of structure after being corroded [19–25], at the same time, many studies have investigated chloride diffusion behavior in multiple phases of normal concrete by numerical simulations in mesoscopic level [26–30]. The HPC has the high amounts of cementitious material and low water–cement ratio [31], the chloride ion diffusion process in the HPC will be different with the normal concrete. However, the mesoscopic level research for the chloride ion diffusion process of HPC is few, scholars mainly focus on the material behavior of the HPC using mesoscopic model. Guo et al. [31] proposed a two-dimensional mesoscopic lattice model of high-performance concrete which accounts for fatigue damage. Huespe et al. [32] investigated the failure mode of high-performance fiber reinforced concrete composites (HPFRC), through proposing a finite element methodology. Forquin et al. [33] developed a mesoscopic numerical model for the ultra-high performance fiber reinforced concrete to investigate the dynamic tension behavior. Feng et al. [34] investigated the mechanical compression properties of UHPC through a 3D mesoscale model. However, the mesoscopic study of the chloride ion diffusion process of HPC containing FA and SF is limited.

In order to investigate the influence of different content combinations of FA and SF on the chloride permeability on HPC, in the present study, the authors carried out a chloride ion diffusion experiment for the HPC. The influence of W/B, binary (PC-FA and PC-SF) and ternary (PC-FA-SF) combinations on the concrete compressive strength and chloride ion diffusion was investigated. The chloride ion diffusion coefficient was modified and the chloride ion diffusion model of HPC was revised based on Fick's second law considering the W/B, temperature, humidity and time. The number simulation of chloride ion diffusion at the mesocopic level for HPC with FA and SF was carried out, and the influence of aggregate shape, distribution, size of HPC on the chloride ion diffusion is investigated.

## 2. Experimental Programs

### 2.1. Detail of Specimens

All specimens were divided into two groups in the experiment. Group I was mainly designed for investigating the influence of W/B, compressive strength on the chloride ion diffusion behavior in the HPC. Group II was mainly designed for analyzing the influence of different mixing methods of mineral admixtures on the chloride ion diffusion behavior.

Twenty-four standard cubic specimens with dimension of 150 mm × 150 mm × 150 mm are made in the group I as shown in Table 1. Eight mix proportions were designed, three specimens were made for every mix proportion. The mix proportions are shown in Table 1. The HPC80 and HPC100 adopted P52.5 ordinary Portland cement, others adopted P42.5 ordinary Portland cement. The coarse aggregate adopted macadam, and the max particle size was 20 mm continuous grading. The fine aggregate adopted fine sand, and the fineness modulus was from 2.5 to 3. The water-reducing rate of the water reducer was from 20% to 25%. The slump of concrete was controlled from 18 cm to 20 cm.

There are twelve specimens in the group II as shown in Table 2. The cement variety and the content of aggregate were the same for all specimens. The specimens A-60 were the HPC without any mineral admixtures. The specimens FA-10, FA-20 and FA-30 were mixed with 10%, 20% and 30% (in weight) FA, respectively. The specimens SF-5 and SF-10 were mixed with 5% and 10% silica fume, respectively. Other specimens were mixed with both FA and SF, the contents of mineral admixtures are shown in Table 2.

**Table 1.** Group I specimens mix proportion.

| Specimen | Water (kg/m³) | Cement (kg/m³) | FA (kg/m³) | SF (kg/m³) | Fine Aggregate (kg/m³) | Coarse Aggregate (kg/m³) | Super Plasticizer (kg/m³) | FA Ratio | SF Ratio | W/B |
|---|---|---|---|---|---|---|---|---|---|---|
| NC50 [1] | 136.35 | 396 | 0 | 0 | 837.4 | 964.25 | 7.13 | — | — | 0.34 |
| NC60 | 122.27 | 444 | 0 | 0 | 791.82 | 1008.91 | 8.44 | — | — | 0.28 |
| NC70 | 108.145 | 488 | 0 | 0 | 747.3 | 1052.56 | 9.76 | — | — | 0.22 |
| HPC50 [2] | 123.135 | 396 | 64 | 35 | 795 | 1005.86 | 8.91 | 13% | 7% | 0.25 |
| HPC60 | 119.63 | 409 | 66 | 36 | 783.34 | 1017.03 | 9.71 | 13% | 7% | 0.23 |
| HPC70 | 115.215 | 421 | 69 | 37 | 769.56 | 1030.22 | 10.54 | 13% | 7% | 0.22 |
| HPC80 | 144 | 501 | 81 | 44 | 647 | 1096 | 8.76 | 13% | 7% | 0.21 |
| HPC100 | 129 | 516 | 84 | 45 | 607 | 1137 | 9.03 | 13% | 7% | 0.20 |

Note: [1] Normal concrete. [2] High performance concrete.

**Table 2.** Group II specimens mix proportion.

| Specimen | Water (kg/m³) | Cement (kg/m³) | FA (kg/m³) | SF (kg/m³) | Fine Aggregate (kg/m³) | Coarse Aggregate (kg/m³) | Super Plasticizer (kg/m³) | FA Ratio | SF Ratio | W/B |
|---|---|---|---|---|---|---|---|---|---|---|
| A-60 | 97.6 | 488 | 0 | 0 | 745 | 1044.5 | 10.5 | — | — | 0.2 |
| FA-10 | 97.6 | 439.2 | 48.8 | 0 | 745 | 1044.5 | 10.5 | 10% | — | 0.2 |
| FA-20 | 97.6 | 390.5 | 97.6 | 0 | 745 | 1044.5 | 10.5 | 20% | — | 0.2 |
| FA-30 | 97.6 | 341.3 | 147 | 0 | 745 | 1044.5 | 10.5 | 30% | — | 0.2 |
| SF-5 | 97.6 | 463.2 | 0 | 24.6 | 745 | 1044.5 | 10.5 | — | 5% | 0.2 |
| SF-10 | 97.6 | 439.5 | 0 | 49.2 | 745 | 1044.5 | 10.5 | — | 10% | 0.2 |
| SF-15 | 97.6 | 414.2 | 0 | 73.8 | 745 | 1044.5 | 10.5 | — | 15% | 0.2 |
| F10-S5 | 97.6 | 415.0 | 48.8 | 24.4 | 745 | 1044.5 | 10.5 | 10% | 5% | 0.2 |
| F20-S5 | 97.6 | 366.2 | 97.8 | 24.3 | 745 | 1044.5 | 10.5 | 20% | 5% | 0.2 |
| F30-S5 | 97.6 | 316.8 | 146.8 | 24.6 | 745 | 1044.5 | 10.5 | 30% | 5% | 0.2 |
| F10-S10 | 97.6 | 390.3 | 48.9 | 48.8 | 745 | 1044.5 | 10.5 | 10% | 10% | 0.2 |
| F20-S10 | 97.6 | 341.3 | 98.2 | 48.5 | 745 | 1044.5 | 10.5 | 20% | 10% | 0.2 |
| F30-S10 | 97.6 | 292.7 | 146.7 | 48.6 | 745 | 1044.5 | 10.5 | 30% | 10% | 0.2 |

### 2.2. Compressive Strength Test

The specimens were made using the standard maintenance method according to the Code for design of concrete structures (GT 50010-2010) [35]. All specimens were de-molded after 24 h and then cured in a standard curing chamber at 20 ± 2 °C and 97–100% relative humidity for 28 days. After curing, the standard cube compressive strength test was carried out to obtain the cube compressive strength.

### 2.3. Chloride Penetration Test

As shown in Figure 1, five surfaces (②–⑥ surfaces) of the specimens were painted using epoxy resin, and the last surface (① surface) was an erosion face. The process of painting epoxy resin is shown in Figure 1. All of the specimens were immersed in a sodium chloride solution with concentration of 3.5% (in weight) for two months, as shown in Figure 2a, the environmental temperature was 25. After that, the specimens were removed from the solution pool and dried naturally for seven days. An electric drill was used to obtain the concrete powders, as shown in Figure 2b. The drilling hole, 16 mm in diameter,

was located in the center of specimen surface. The concrete powders were drilled with an interval from 5 mm to 30 mm away from the exposed surface (i.e., 5 mm, 10 mm, 15 mm, 20 mm, 25 mm and 30 mm depths from the surface). The chloride concentration of the concrete powders was determined by using the Rapid Chloride Test (RCT)-500 system produced by Germann Instruments company in Copenhagen, Denmark, as shown in Figure 2c. The RCT method was verified as having a good correlation with these methods recommended by ASTM C1202 [36] and AASHTOT260-97 [37].

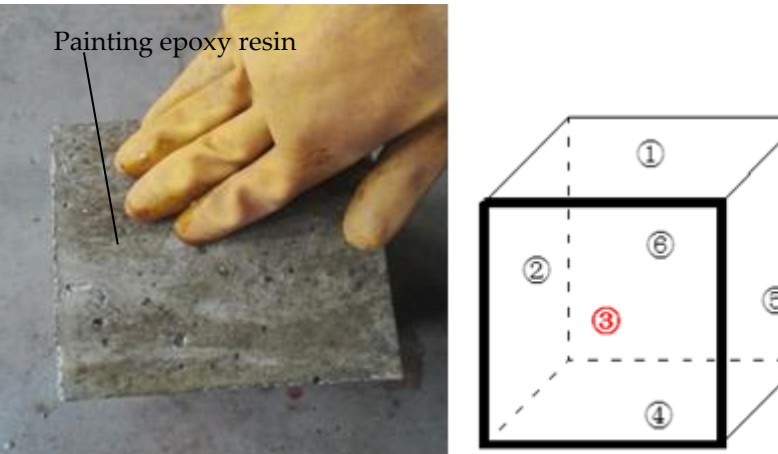

**Figure 1.** Process of painting epoxy resin.

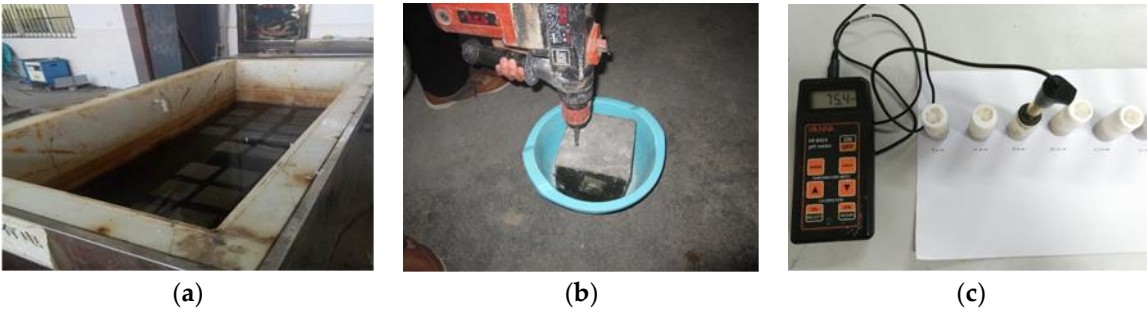

**Figure 2.** Chloride penetration test set-ups: (**a**) specimens immersing; (**b**) drilling holes; (**c**) RCT test.

## 3. Experimental Result Analysis

The influence of the W/B and the mixing method of FA and SF on the compressive strength and chloride concentration of HPC is discussed through experimental results.

### *3.1. Influence of the W/B*

#### 3.1.1. Concrete Compressive Strength

Figure 3 shows the compressive strength of Group I specimens, the value in the bracket is the W/B. As seen in the Figure 3, no matter normal concrete or the HPC, the compressive strength of specimens increases with the decrease in the W/B. For the normal concrete, the W/B decreases 35.3% from 0.34 to 0.22, the concrete strength increases 47.6% from 47.59 MPa to 70.26 MPa. For the HPC, the W/B decreases 20% from 0.25 to 0.2, the concrete strength increases 95.9% from 54.08 MPa to 105.98 MPa. The ratio of W/B deceasing value and strength increasing value for normal concrete is 0.74, but for the HPC is 0.20, so the influence of the W/B on the concrete strength for normal concrete is obviously more than the HPC.

Sengul and Tasdemir [7], Chindaprasirt et al. [38] present that the compressive strength of concrete with mineral admixtures is smaller than the normal Portland concrete under the same W/B. This result also can be verified by the compressive strength of specimens

OC70 and HPC70. However, the compressive strength of specimens HPC 60 and HPC50 is 3.5% and 13.5% larger than specimens OC60 and OC50, respectively. This can be attributed to the smaller W/B.

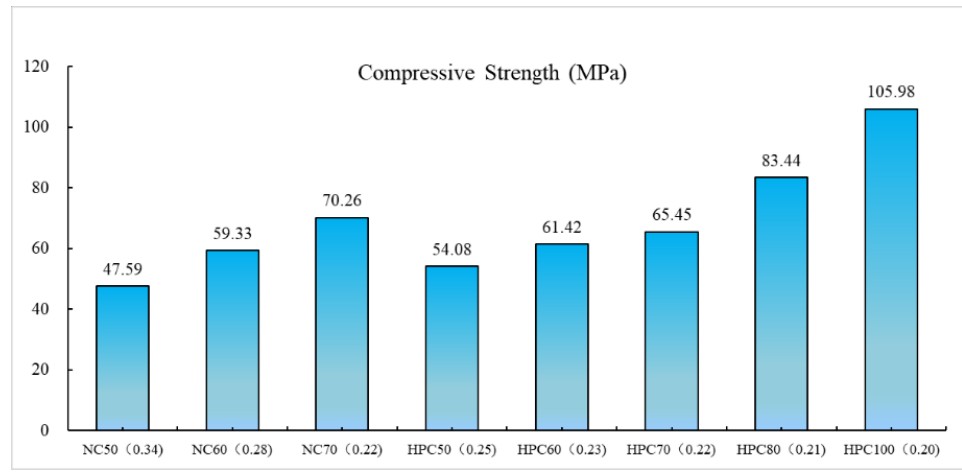

**Figure 3.** Compressive strength of the Group I specimens.

3.1.2. Chloride Ion Concentration (Permeability)

The W/B is an important factor impacting the chloride ion diffusion in the HPC. The chloride concentrations of part specimens were obtained as shown in Figure 4. For both the normal concrete and the HPC, from a depth of 7.5 mm to 17.5 mm, the influence of W/B on the chloride ion concentration is most significant; under the same diffusion depth, the chloride concentration decreases with the increase in W/B. The lower W/B reduces the amount of capillary pores and subsequently affects the resistance of concrete against chloride transport [6]. The concentration gradient of chloride ion is gradually decreased in the deeper regions of the concrete, and especially for depths beyond 22.5 mm, the influence of the W/B is very small.

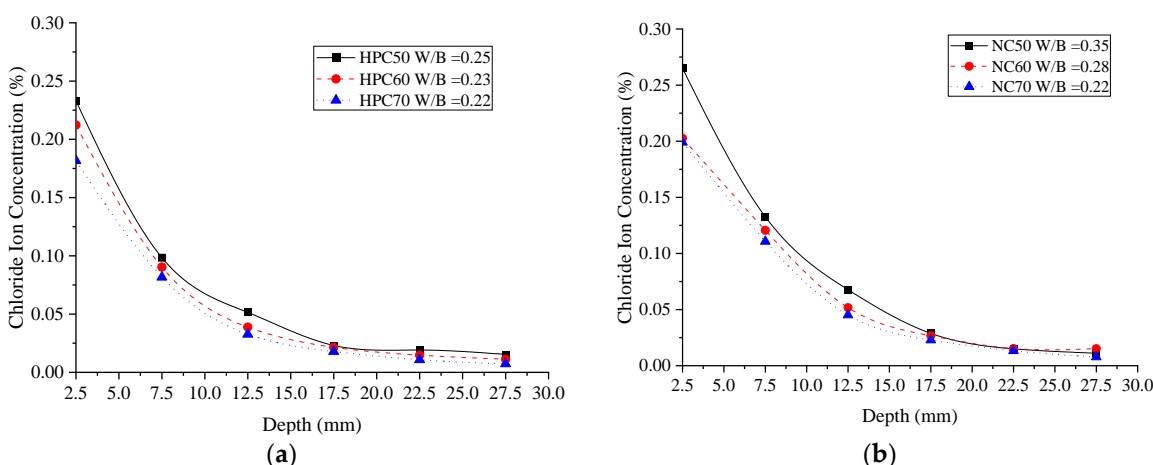

**Figure 4.** Chloride ion concentration of the specimens: (**a**) Normal concrete; (**b**) HPC.

Figure 5 shows the chloride ion concentration distribution of the normal concrete and HPC. From depth 0 mm to 7.5 mm, the chloride ion concentration of HPC decreases substantially compared to the normal concrete, the chloride ion concentration of specimen HPC 100 is about 16% of the specimen NC50 at depth 7.5 mm. The pozzolanic reaction is relatively slow compared to the hydration of Portland cement, which is the main reason of high early age permeability of the concretes containing pozzolanic materials. At later ages, however, the pozzolanic materials can be more effective in improving concrete properties

due to the pozzolanic reaction continuing at a higher rate for a longer period, so the chloride ion concentration of the HPC decreases slower than the normal concrete from depths of 7.5 mm to 17.5 mm.

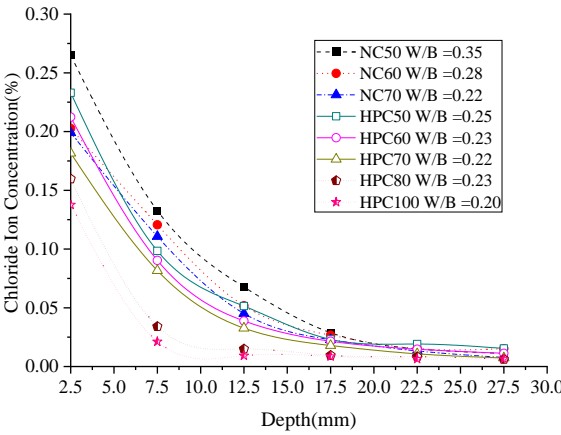

**Figure 5.** Chloride concentration distribution of Group I specimens.

### 3.2. Influence of the Mixing Method of Mineral Admixtures

### 3.2.1. Concrete Compressive Strength

The concrete compressive strengths of Group II specimens are presented in Figure 6. For binary combination specimens SF5, SF10 and SF15, the concrete compressive strength increases 11.93% when the content of the silica fume increases from 5% to 15%. However, the concrete compressive strength of specimen SF15 is 5.69% smaller than specimen SF10. The same situation appears for the compressive strength of specimens FA10, FA20 and FA30. The experimental results conclude that the contents of the SF or FA developing the strength is limited. Figure 7 shows the concrete compressive strength of specimens with different FA content obtained by Jerath and Hanson [39], which also concludes that the concrete compressive strength at the same age firstly increases, then decreases with the FA content increases.

Interestingly, it can be found from the strength of specimens F10-S5 to F30-S5 and F10-S10 to F30-S10, the compressive strength of specimens F10-S5 to F30-S5 decreases from 8.9% to 11.5% with the increase in FA when the content of the silica fume keeps unchanged. The compressive strength of specimens F10-S10 to F30-S10 decreases from 5.6% to 22.2% with the increase in SF when the content of the FA keeps unchanged.

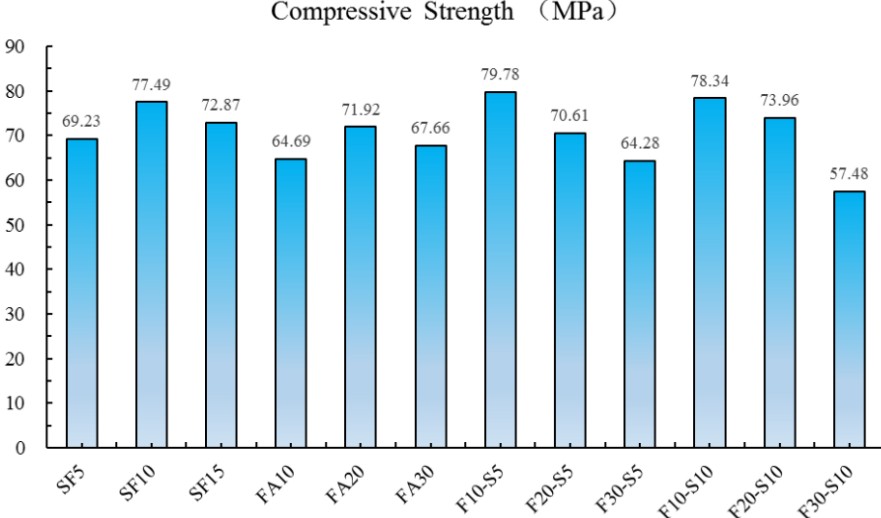

**Figure 6.** Compressive strength of the Group II specimens.

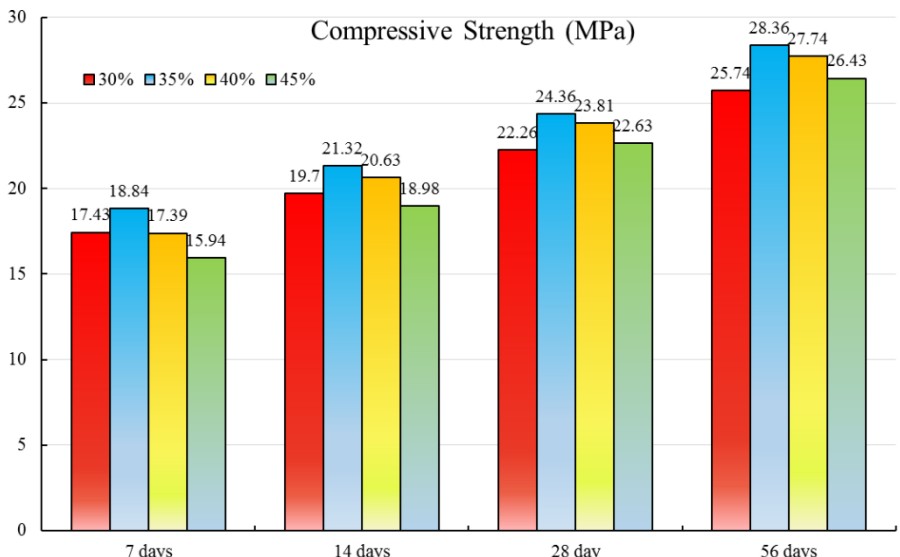

**Figure 7.** Compressive strength of the specimens in Jerath and Hanson's test.

3.2.2. Chloride Ion Concentration

It has been shown in laboratory and field studies that FA can mitigate corrosion by hindering chloride penetration as well as decreasing the content of free chlorides. The use of FA improves the distribution of pore size and pore shape of concrete, and the presence of $C_3A$ in FA can absorb more chloride ions to form Friedel's salt ($C_3A \cdot CaCl_2 \cdot 10H_2O$) [40]. Moreover, fly ash reacts pozzolanically with calcium hydroxide to produce calcium-silicate hydrate (C-S-H), which absorbs more chloride ions and block the ingress path [9,40]. The SF is also more efficient in prohibiting chloride ingress inside the HPC as well as at the surface layer than round granulated blast furnace slag and metakaolin [6]. In this study, utilizing the specimens in Table 2, the influences of binary blends (Portland cement–fly ash (PC-FA) and Portland cement– silica fume (PC-SF)) and ternary blends (Portland cement–fly ash–silica fume (PC-FA-SF)) on the chloride permeability are investigated: the experimental results are shown in the Figures 8 and 9. In Figure 8, the chloride ion concentration of the specimens with FA or SF is smaller than normal concrete when the diffusion depth is smaller than 17.5 mm, but the chloride concentrations have no obvious difference beyond 17.5 mm. As shown in Figure 8a), the chloride ion concentration decreases with the FA increases from depth 2.5 mm to 12.5 mm. A similar situation also appears in Figure 8b: the chloride ion concentration decreases with the growth of the SF content from a depth of 2.5 mm to 7.5 mm.

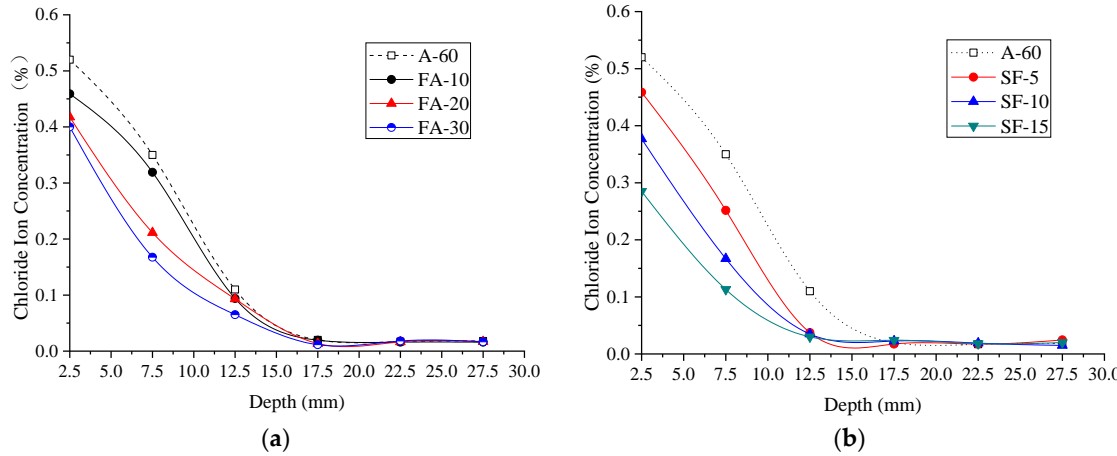

**Figure 8.** Chloride ion concentration of specimens with binary blends: (**a**) Portland cement–fly ash; (**b**) Portland cement–silica fume.

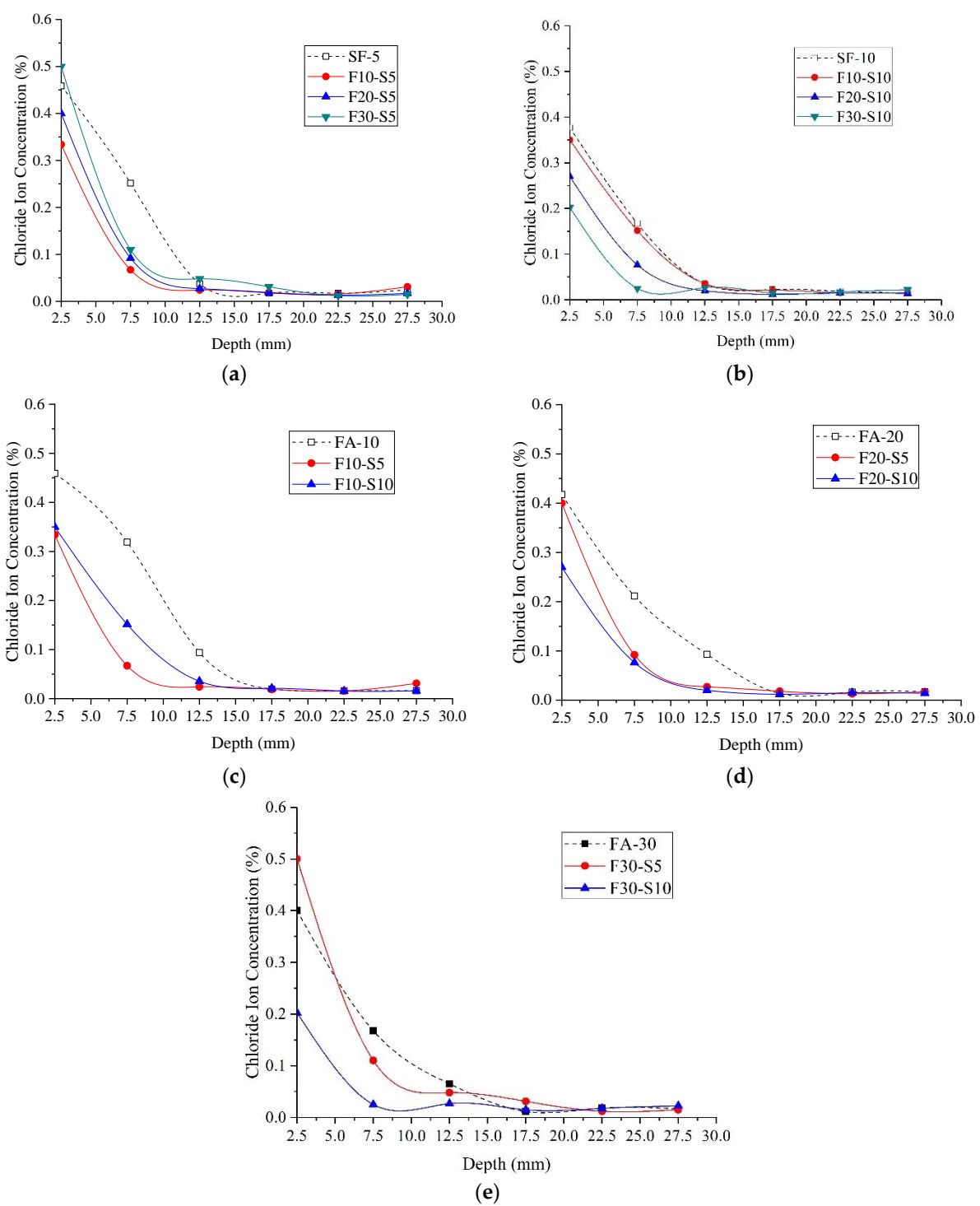

**Figure 9.** Chloride ion concentration of specimens with binary and ternary blends: (**a**,**b**) Portland cement–silica fume with Portland cement–fly ash–silica fume; (**c**–**e**) Portland cement–fly ash and Portland cement–fly ash–silica fume.

The profiles of chloride ion concentration of specimens with ternary blends are shown in Figure 9. When the diffusion depth is less than 12.5 mm (In Figure 9a,b,e) or less than or equal to 12.5 mm (In Figure 9c,d), the chloride ion concentration of specimens with ternary blends is smaller than specimens with binary blends under the same depth, which concludes that the ternary combination of PC-FA-SF has more efficient in prohibiting chloride's ingress inside the specimens than the binary combination of PC-FA or PC-SF.

Figure 9a presents the chloride ion concentration of specimens with different contents of FA and 5% SF. Keeping the silica fume content unchanged, the order of chloride ion concentration is F10-S5 < F20-S5 < F30-S5. When the depth is 12.5 mm, the chloride ion concentration of specimen F10-S5 is 50% less than specimen F30-S5. This situation is different from the chloride ion concentration distribution for the specimen containing only FA or SF. Interestingly, in Figure 9b, when the content of SF increases to 10%, the chloride ion concentration decreases with the FA increases under the same depth. In Figure 9c–e, the similar situation above is obviously more. When the content of FA is 10% (Figure 9c), the chloride ion concentration decreases with the SF increases. However, when the content of FA is 20% and 30% (Figure 9d,e), the chloride ion concentration decreases as the silica fume increases when the depth less than 17.5 mm. The above situation is attributed to the total content of mineral admixtures increasing, which leads to diffusion resistance improvement for improving the density of concrete.

## 4. Modification of the Chloride Ion Diffusion Model

### 4.1. Ideal Model-Fick's Second Law

Buenfeld et al. [41] considered that the progress of chloride transportation in the concrete is the unsteady diffusion process. For Fick's second law, in unit time and unit volume, the mass of material flowing out is different from that flowing into the internal structure of concrete, and the concentration of any point within it varies with time and space; this progress is also the unsteady process. Many scholars [42,43] concluded that the chloride ion diffusion process can be described by Fick's second law:

$$\frac{\partial C}{\partial t} = D \frac{\partial^2 C}{\partial x^2} \tag{1}$$

When the boundary condition is: $C(0, t) = C_s$, $C(\infty, t) = C_0$; the origin condition is $C(x,0) = C_0$, above equation can be solved as:

$$C(x,t) = C_0 + (C_s - C_0) \left[ 1 - erf \left( \frac{x}{2\sqrt{Dt}} \right) \right] \tag{2}$$

where $C(x, t)$ is the chloride ion concentration of the inner part concrete at time $t$ and distance $x$, $C_0$ is the chloride ion origin concentration, $C_s$ is the surface chloride ion concentration, $D$ is the diffusion coefficient, $erf(z)$ is error function, $erf(z) = 2/\sqrt{\pi} \int_0^\pi e^{-z^2} dz$.

Fick's second law is an ideal model, which sets three suppositions: (1) the concrete is a semi-infinite homogeneous material; (2) the chloride ion does not react with concrete during the diffusion process; and (3) the diffusion coefficient is the constant. However, many studies have already concluded that: firstly, the concrete has micro-cracks and a void pore, and it is not a completely homogeneous material. Secondly, a chloride ion will bind with the cementitious paste and form Friedel's salt. At last, the chloride ion diffusion coefficient is not constant but time-dependent. The structure of HPC is more complicated than normal concrete because of the mineral admixture, the chloride diffusion process is affected by many factors. In this study, the diffusion coefficient was modified considering the influence of the water–bind ratio, time, temperature, humidity and deterioration effect of concrete, and a modified model is proposed.

### 4.2. Modified Diffusion Model

#### 4.2.1. Relationship between the Diffusion Coefficient and W/B

It was reported that the diffusion coefficient was time-dependent, and it was affected by W/B, temperature, humidity, the combination affectation between concrete and chloride ion, etc. Due to having a different composition compared with normal concrete, the diffusion coefficient should be modified. As shown in Figure 10, the normalized chloride diffusion coefficients of the HPC were obtained in this experiment and research results from

experimental results obtained by Chalee and Jaturapitakkul [8], Moffat et al. [9], Thomas and Matthews [44] and Shafq [45].

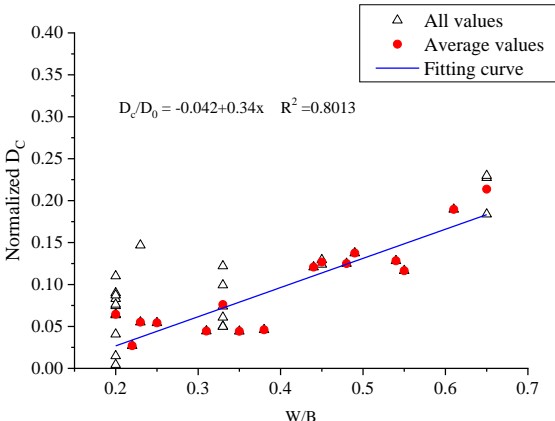

**Figure 10.** Chloride diffusion coefficient of concrete after different W/B.

Some scatters exist in the calculated chloride diffusion coefficients at different depths for each specimen. This could be attributed to the measurement errors of chloride concentration during the experiment. The corresponding average values, however, have relatively small scatters for the specimens with the same W/B. It can be shown in Figure 10 that the chloride diffusion coefficient shows the linear relationship with W/B, the diffusion coefficient of normal concrete is from 4.7 times to 12 times as compared with the HPC when the W/B is from 0.2 to 0.65. A coefficient is also developed to consider the W/B on the chloride diffusion coefficient based on the experimental data, which is expressed as:

$$R_C = \frac{D_C}{D_0} = -0.042 + 0.34 \mathrm{W/B} \tag{3}$$

where $R_C$ is the coefficient considering the effect of the W/B, which is the diffusion coefficient ratio of HPC to normal concrete. $D_C$ and $D_0$ are the diffusion coefficient of HPC and normal concrete, respectively.

### 4.2.2. Relationship between the Diffusion Coefficient and Time, Temperature

The relationship between chloride ion diffusion coefficient $D(t, T)$ and time, temperature can be presented from the experimental results obtained by Mangat and Molloy [46]. The experimental results showed that the time and temperature effect should be considered to more accurate of chloride diffusion coefficient in concrete, so the chloride diffusion coefficient can be expressed as:

$$D(t, T) = D_0 f_t f_{T_b} \tag{4}$$

$$f_t = \left( \frac{t_{\exp}}{t - t_{build}} \right)^m \tag{5}$$

where $D(t, T)$ was the chloride diffusion coefficient for the real structures, $f_{T_b}$ was the temperature correction coefficient. $f_t$ was the time correction factor, which can be expressed by Equation (5). $t_{\exp}$ was the experimental total time, $t$ was the calculated time, $t_{build}$ was the construction completion time. $m$ was the experimental parameter, which was obtained based on the different kind of cement. For normal Portland cement, m was in the range of 0.2–0.3, and for HPC with fly ash and slag, the higher values (0.5–0.7) could be used [41].

The chloride diffusion coefficient should also consider the temperature effect, which can be presented as [47]:

$$f_{T_b} = \exp \left[ \frac{U}{R} \left( \frac{1}{T_{28}} - \frac{1}{T_b} \right) \right] \tag{6}$$

where $U$ was the activated energy during the diffusion process of chloride ion in concrete and was taken as 35,000 J/mol, $R$ was the gas constant and taken as 8.314 J/(mol.k), $T_{28}$ was the absolute temperature corresponding to a curing of 28 days and was generally taken as 293 K, and $T_b$ was the average absolute temperature related to calculation.

### 4.2.3. Relationship between the Diffusion Coefficient and Humidity

Bitaraf and Ammadis [48] pointed out that the transport of chloride ions in concrete must take pore water as continuous medium. The more pore water and the larger the section size, the smoother the diffusion of chloride ions would be. Therefore, the water content or saturation of concrete determined the number of chloride ion transport channels. At the same time, the relative humidity could be used to express the water content or saturation of concrete. When the water content inside concrete was relatively low, the chloride ion diffusion channel decreased, and its diffusion speed slowed down accordingly. Therefore, the humidity is also a factor that affects the diffusion of chloride ions: the relationship between chloride ion diffusion coefficient and humidity could be expressed as:

$$D_h = D_0 \left[ 1 + \frac{(1-h)^4}{(1-h_c)^4} \right]^{-1} \tag{7}$$

where $D_h$ is the diffusion coefficient of chloride ion when the humidity is $h$, $h_c$ is the critical relative humidity, and the value is 75% at the atmospheric environment, 90% at the splash zone and 95% at the tidal range zone.

In conclusion, the model of diffusion coefficient of chloride ion considering water–cement ratio, time, temperature, and humidity can be shown

$$D(t) = R_C D_0 \exp\left[ \frac{U}{R} \left( \frac{1}{T_{28}} - \frac{1}{T} \right) \right] \left[ 1 + \frac{(1-h)^4}{(1-h_c)^4} \right]^{-1} \left( \frac{t_{\exp}}{t - t_{build}} \right)^m \tag{8}$$

### 4.2.4. Initial Value Condition

The pouring and curing process of concrete is not eroded by chloride ion, so the chloride content inside the concrete is zero:

$$C_0(x,0) = 0 \tag{9}$$

### 4.2.5. Boundary Condition

The surface chloride ion concentration of concrete is not the fixed value, but changes with time. Liu et al. [49] proposed that the relationship between surface chloride ion concentration and time can be obtained using followed equation:

$$C_s(t) = C_{ref}\left(1 - e^{-at}\right) \tag{10}$$

where $C_{ref}$ was surface chloride concentration at the reference time $t$, and $\alpha$ was the regression parameter.

## 5. Random Aggregate Model of Concrete

### 5.1. Basic Theory

In aggregate gradation theory, the particle size and the gradation are the most important factors. In existing research, the definition of aggregate particle size is still difficult to unify and lack of practicability. Therefore, the sieving method is used to determine aggregate particle size, and the grain size can be calculated based on the upper and lower limits.

Aggregate gradation includes two aspects: particle size and the relative number of particles in each gradation. The sieving curve is used to determine aggregate gradation. Although this method can obtain the size and distribution of aggregate, it can be more

vivid through numerical characterization. Commonly the modulus of fineness and specific surface area are the numerical indicators.

The existing research results show that the concrete gradation theory can be divided into two types: continuous gradation and discontinuous gradation. If the content of aggregates with different particle sizes is close to reasonable continuous gradation curve, the strength and maximum compactness of the concrete will reach the best effect according to this gradation. In this paper, the Walraven's model [50] is used to describe the aggregate gradation, which translate the spatial spherical aggregate to the flat round aggregate based on the Fuller curve. Based on the method of probability statistics, Walraven used the cumulative distribution function to transform the three-dimensional gradation curve into the probability that an aggregate diameter $D < D_0$ for any point on an internal section, where the probability can be shown as:

$$
\begin{aligned}
P_c(D < D_0) = P_k[1.065(D_0/D_{\max})^{0.5} - 0.053(D_0/D_{\max})^4 - \\
0.012(D_0/D_{\max})^6 - 0.0045(D_0/D_{\max})^8 - 0.0025(D_0/D_{\max})^{10}]
\end{aligned}
\tag{11}
$$

where $D_0$ was sieve diameter, $D_{\max}$ was maximum aggregate particle size, $P_k$ was the percentage of aggregate volume in the total volume, the value was 0.75. The probability $P_c$ of the different aggregate gradation of the two-dimensional section of the specimen could be obtained by replacing the boundary value of the three-dimensional aggregate content $P_K$ with the diameter of the aggregate at all levels in formula (11), and then the two-dimensional gradation curve could be obtained.

### 5.1.1. Circular Aggregate Generation and Release Process

The release process of circular aggregate is introduced in this paper firstly. The radius of the circular aggregate is assumed to be $R_i$, and its corresponding circular coordinate is $(x_i, y_i)$, then the calculation method of the geometric model of the aggregate is as follows:

(1) Defining the radius and the centroid coordinates of aggregate $(x_i, y_i)$ The $x_i$ and $y_i$ need to satisfy the following requirements: $x_i = \frac{d}{2} + (a - d) \cdot \eta$, $y_i = \frac{d}{2} + (b - d) \cdot \eta$. Where, $a$ and $b$ are the side length of release area, respectively, $a = b = 150$ mm in this paper. $\eta$ is a random number between 0 and 1.

(2) Overlapping judgment. This process is determining whether the generated $i$ aggregate overlaps with the generate $i - 1$ aggregate. If un-overlapped, the area of the aggregate is calculated and the information of the aggregate is recorded; if overlapped, turning back to the step (1) and recoding the value. The basis of overlap judgment is: If the interval length $d$ between two circular aggregate centroid is larger than the sum of their radius, namely $d > R1 + R2$, it can be concluded that the circular aggregate with radius $R_1$ does not overlap with the circular aggregate with radius $R_2$.

(3) Judging of the aggregate area. If the condition $\Sigma A_i \geq A_0 \cdot a \cdot b$ is not met, the researcher should be turn back to step (1) and continue to release. If the above condition is met, the aggregate is finished. Where $A_0$ is the area fraction of the calculated aggregate.

### 5.1.2. Polygonal Aggregate Generation and Release Process

In this paper, the generation process of polygonal random aggregate is realized as follows: Firstly, based on the simulation of circular aggregate, the diameter of random circle is assumed as the particle size of polygonal aggregate, and the location and size of the aggregate are determined by using the randomly generated circle. Secondly, a polygonal base aggregate is generated randomly in the random circle, and the area of the polygon is greater than or equal to 0.75~1.1 times of the area of the random circle. Finally, the longest edge is set as the reference edge, the polygonal random aggregate model is established by using the rule that the reference edge generates new points and determines whether the new points meet the outward extension condition.

### 5.1.3. Examples of Random Aggregate

The content of aggregate in each tested block was calculated based on the different mix proportion of test specimens in Table 1 and then the geometric model of circular aggregate was generated by MATLAB 2021a programming language according to the random number theory using the Monte Carlo method.

For a circular aggregate, the area *S* (volume fraction of aggregate) of a certain particle size in the two-dimensional model is calculated according to Equation (11) according to the size and grade of aggregate particle size, and then the amount of aggregate corresponding to each diameter is further determined. In the section of the specimen, the position of randomly generated aggregate centroid was firstly determined, and then circular aggregate is randomly generated according to the corresponding aggregate diameter at that position. Finally, the aggregate of all particle sizes is released, that is, the establishment of circular random aggregate model was completed

The mixing ratio of each specimen is shown in Table 1. According to the mixing ratio of each specimen in test, the volume percentage ratio of aggregate to concrete $P_k$ can be obtained, as shown in Table 3.

**Table 3.** Proportion of aggregate volume to concrete for each strength grade.

| Specimen | Mass of Aggregate | Volume of Concrete | Volume of Aggregate | Proportion of Aggregate Volume |
|:---:|:---:|:---:|:---:|:---:|
| NC50 | 6.08056875 | 0.003375 | 0.002252063 | 0.67 |
| NC60 | 6.07746375 | 0.003375 | 0.002250913 | 0.67 |
| NC70 | 6.074510625 | 0.003375 | 0.002249819 | 0.67 |
| HPC50 | 6.077919375 | 0.003375 | 0.002251081 | 0.67 |
| HPC60 | 6.07624875 | 0.003375 | 0.002250463 | 0.67 |
| HPC70 | 6.074274375 | 0.003375 | 0.002249731 | 0.67 |
| HPC80 | 5.882625 | 0.003375 | 0.00217875 | 0.65 |
| HPC100 | 5.886 | 0.003375 | 0.00218 | 0.65 |

Specimens NC50 and HPC80 are used to build the randomly geometric of circular and polygon aggregates. Table 4 shows the distribution probability of aggregate particle size of specimens NC50 and HPC80. Combining the proportion of aggregate volume with the Equation (11), the probability $P_c$ that aggregate diameter $D$ less than $D_0$ in any cross section is calculated, then the aggregate number for every diameter is also confirmed. Figure 11 show the circle aggregate model of specimen NC50 and HPC80.

**Table 4.** Distribution probability of particle size of each aggregate in the specimens NC50 and HPC80.

| Specimen | $D_0$ | 22 | 18 | 12 | 5 |
|:---:|:---:|:---:|:---:|:---:|:---:|
| NC50 | $D_0/D_{max}$ | 1 | 0.81 | 0.55 | 0.23 |
|  | $P_c$ | 0.65 | 0.61 | 0.51 | 0.33 |
| HPC80 | $D_0/D_{max}$ | 1 | 0.81 | 0.55 | 0.23 |
|  | $P_c$ | 0.63 | 0.59 | 0.49 | 0.32 |

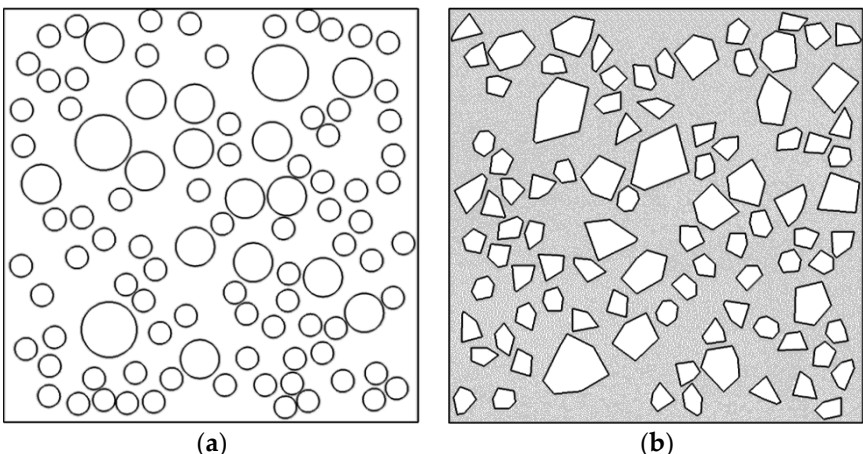

**Figure 11.** Circle aggregate model of specimens: (**a**) NC50; (**b**) HPC80.

## 6. Mesoscopic Simulation of Chloride Ion Diffusion

*Simulation Method*

Software COMSOL Multiphysics 5.1 provides modules for describing and analyzing various physical fields which including chemical transfer, fluid flow, and dynamic and solid mechanics. The Transport of Diluted Species Interface in this software is suitable to simulate the material transfer process under the consideration of convection–diffusion and diffusion processes, so the chloride ion diffusion process in high performance concrete in this paper is simulated using this module. The simulation steps are shown as follows:

i.     Choosing the Transport of Diluted Species Interface in the COMSOL software and importing the two-dimensional concrete random aggregate geometric model proposed in Part 4 into the software and set the cell.
ii.    Defining the parameter value in transfer equation and diffusion coefficient model based on the Equation (11).
iii.   Defining the boundary condition $C_s$, diffusion coefficient and initial value condition $C_0$.
iv.    Dividing the mesh.
v.     Defining a two-dimensional transversal and extracting the chloride ion concentration on the transversal lines of 2.5 mm, 7.5 mm, 12.5 mm, 17.5 mm, 22.5 mm, and 27.5 mm, respectively.

Dividing the Mesh

Of the above steps, dividing the mesh is the most important. In this paper, concrete is regarded as a three-phase composite composed of aggregate, mortar, and the transition layer between them. The COMSOL software is used to carry out the grid-process for the two-dimensional random aggregate model generated in Part 4, and then the two-dimensional random aggregate finite element model is established. According to the research result from Basheer et al. [51], the thickness of transition layer is from 0 μm to 10 μm, in this paper, the thickness of the transition layer in the three-phase random aggregate model is 75 μm. The finite element model is shown in the Figure 12.

The process of mesh generation in COMSOL Multiphysics is defined by mesh sequence, which consists of operation characteristics and attribute characteristics. Among them, operation features are mainly for mesh type, copy mesh, refine mesh, or transform mesh, while attributed features are mainly for mesh type related features such as size, distribution, and proportion. Therefore, when creating a mesh in software, the operation characteristics of the mesh need to be defined first to generate or modify the corresponding geometry of the mesh. Then, the local attribute characteristics is defined. When adding local attribute characteristics in the software, the selected local attribute characteristics will appear on the corresponding sub-nodes under the operation characteristics node, especially when the

selected target is the same. The global attribute node is automatically overwritten by the local attribute feature node.

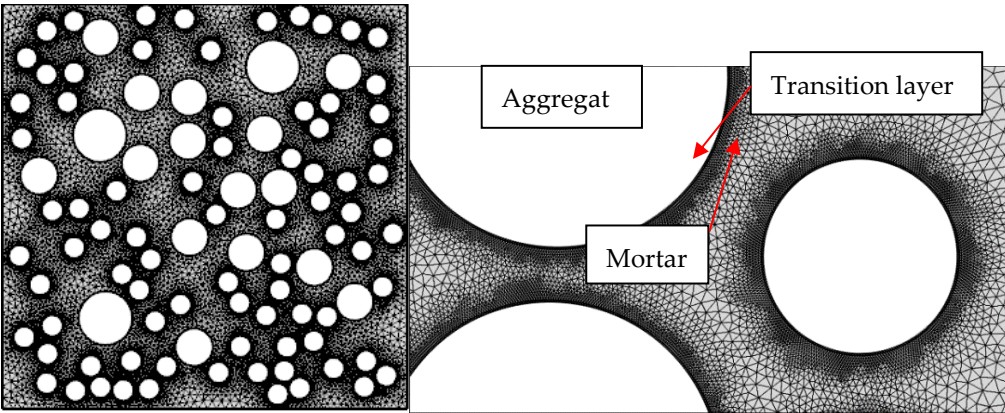

**Figure 12.** Random aggregate finite element model.

## 7. Microscopic Model Validation and Parameters Analysis

### 7.1. Model Validation

In order to verify the reliability and accuracy of the microscopic model in this paper, the measured concentration value in the experiment is compared with simulation value. In the specific test process, due to the presence of chloride crystals on the surface of the specimen and chloride salts in the micro-pores, it is impossible to accurately determine the chloride ion concentration on the concrete surface. Therefore, the chloride ion concentration at 2.5 mm is regarded as the chloride ion degree on the concrete surface, that is, chloride ion begins to diffuse into the concrete interior from 2.5 mm. Figure 13 shows the chloride ion concentration of test and microscopic simulation at different depth. Generally, it can be found from Figure 13 that the simulation and the tested value of the chloride ion concentration under the same depth is close, the average ratio of simulation value to tested value is 0.91. The simulation and tested chloride ion concentration of the specimens HPC60, HPC80, and HPC100 under the depths of 7.5 mm and 12.5 mm is slightly different, it may be the reason that the mortar content is not uniform.

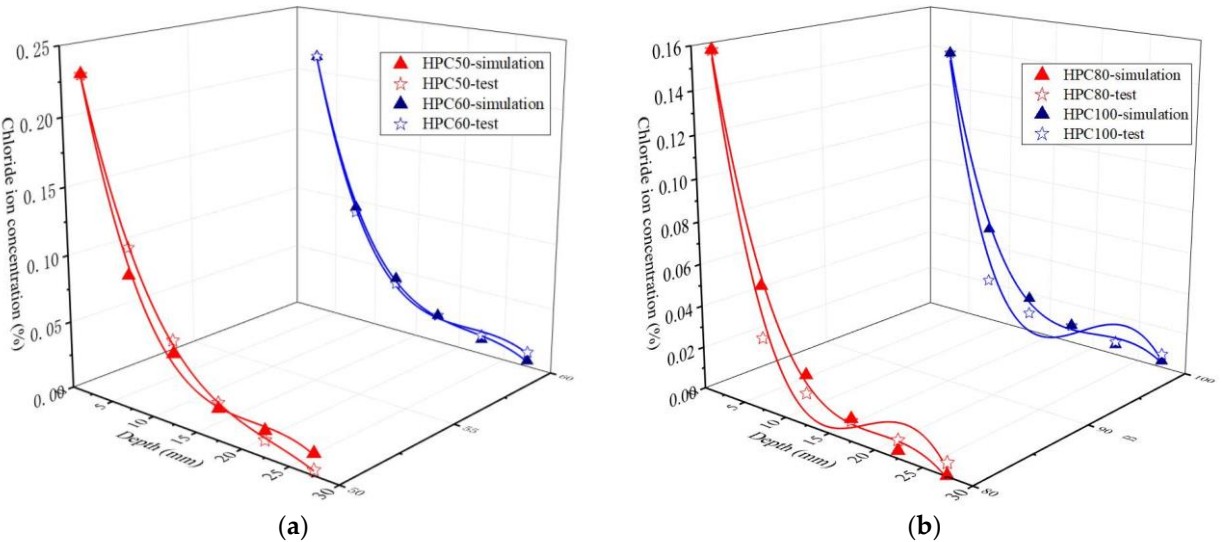

(**a**)  (**b**)

**Figure 13.** Chloride ion concentration of test and microscopic simulation at different depth: (**a**) HPC50 and HPC60; (**b**) HPC80 and HPC100.

### 7.1.1. Effect of Aggregate Shape on Chloride Ion Permeability

In the mesoscopic model, the content, particle size, and position of aggregate are kept the same, and the shape of aggregate is changed to study its influence on the diffusion of chloride ion in concrete. The surface chloride ion concentration is 0.6, the thickness of transition layer is 75 μm, and the diffusion time is 88 days. The numerical simulation results of specimens HPC50 with circle and square aggregate are shown in Figure 14. Figure 14 shows the chloride ion concentration of specimen HPC50 with different aggregate shapes, the chloride ion concentration at the same depth in the Figure 15 is the average value of all chloride ion concentrations at a boundary.

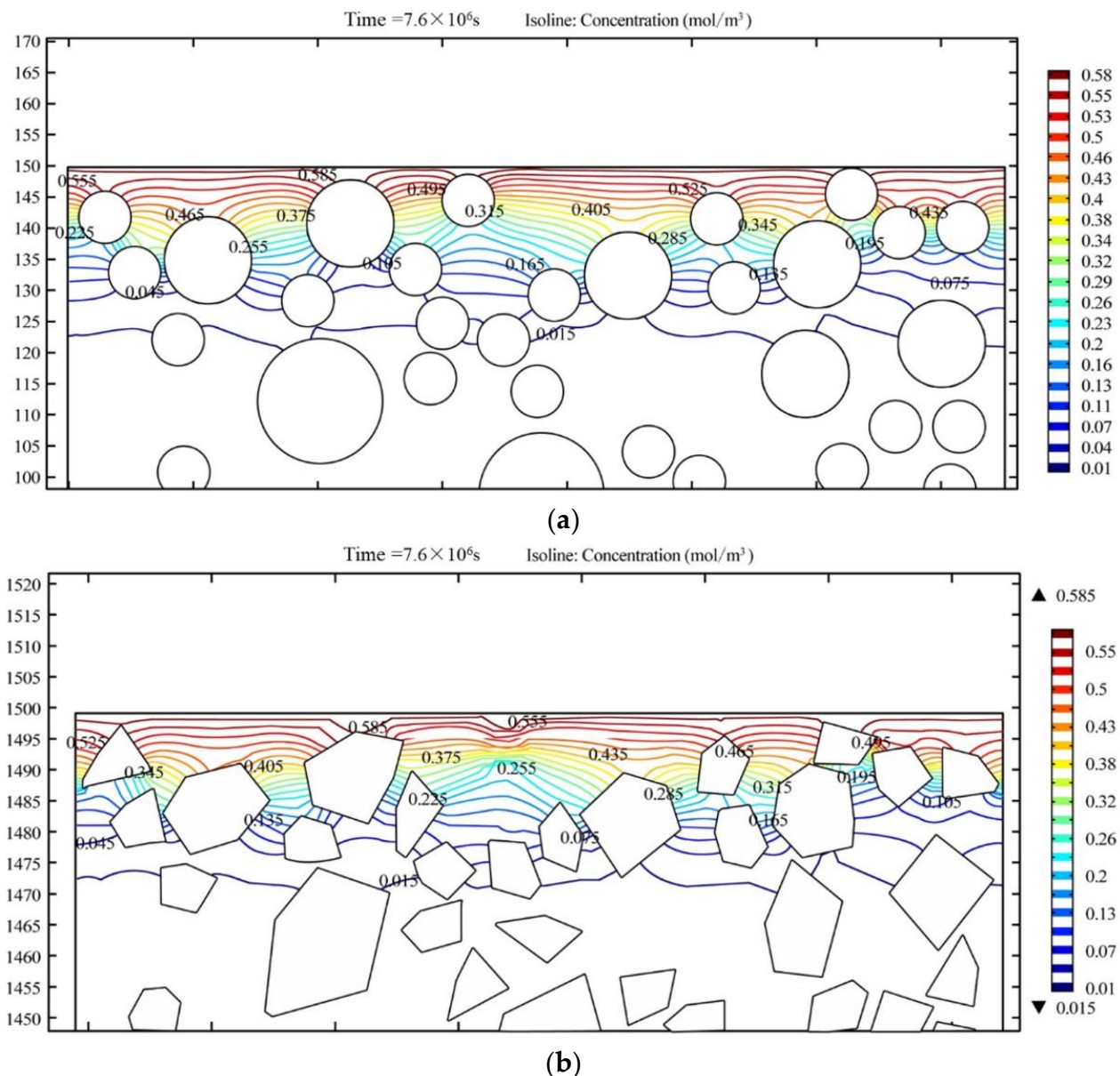

**Figure 14.** Numerical simulation results of specimens HPC 50 with circle and square aggregate: (**a**) Circle aggregate; (**b**) Square aggregate.

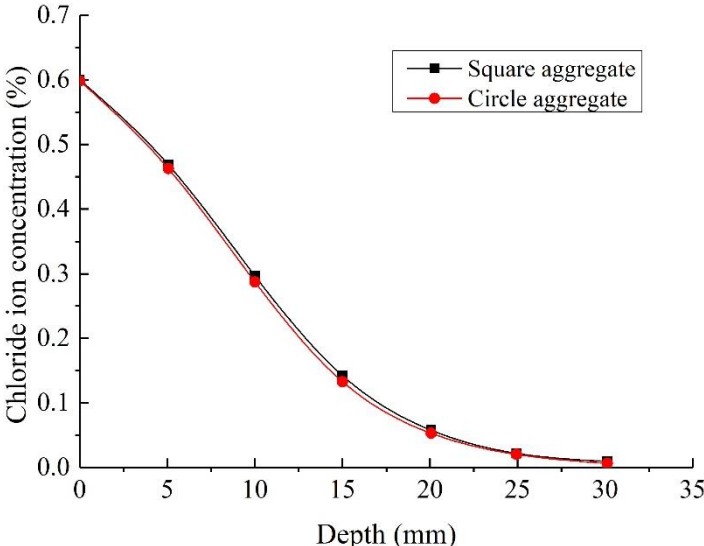

**Figure 15.** Chloride ion concentration of specimen HPC50 with different aggregate shape.

It can be found from Figure 15 that no matter the circle aggregate or square aggregate, the average chloride ion concentration under the depth from 0 mm to 30 mm is close. The average deviation ratio of chloride ion concentration between square aggregate and circle aggregate is 10%. Figure 14 shows that the aggregate shape effects the chloride ion concentration around the aggregate. From the above analysis it can be concluded that the aggregate shape has little influence on the chloride ion diffusion process in the concrete inside. This conclusion is same with normal concrete in Du's study [27]. Du et al. [27] found that aggregate shape has a negligible influence on chloride diffusivity in concrete.

7.1.2. Effect of Aggregate Distribution on Chloride Ion Permeability

Due to the influence of shape of aggregate on the chloride ion diffusion in concrete is not obvious, the circle aggregate is taken as an example in the microscopic model of HPC50. The content and particle size of aggregate are kept the same, and only the distribution of the aggregate is changed to study its influence on the chloride ion diffusion in concrete. The surface chloride ion concentration is still 0.6, the transition layer thickness is still 75 μm, and the erosion time is still 88 days. The numerical simulation results are shown in Figure 16. The average chloride ion concentration of specimen HPC50 with different aggregate distributions under the same depth is presented in Figure 17.

It can be seen from Figure 16 that aggregate only affects the transport path and director of chloride ions in concrete, and has little effect on the average chloride ion concentration at the same depth. Figure 17 also shows that the chloride ion concentrations between the different aggregate diffusion are close except under the 15 and 20 mm. The average deviation ratio of chloride ion concentration between Position 1 and Position 2 is 18.9%. Above finding is also similar with normal concrete in Du et al.'s study [27].

Time =7.6×10⁶s    Isoline: Concentration (mol/m³)

(**a**)

Time =7.6×10⁶s    Isoline: Concentration (mol/m³)

(**b**)

**Figure 16.** Numerical simulation results of specimens HPC 50 with different aggregate distribution: (**a**) Position 1; (**b**) Position 2.

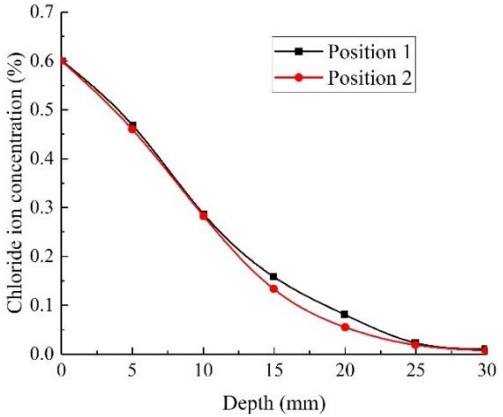

**Figure 17.** Chloride ion concentration of specimen HPC50 with different aggregate distribution.

### 7.1.3. Effect of Aggregate Size on Chloride Ion Permeability

Keeping the content of aggregate unchanged, the influence of aggregate size on the chloride ion concentration has been studied. The aggregate sizes in Figure 18a are 20 mm, 14 mm and 7.5 mm (Size 1), which are 30 mm, 15 mm, and 7.5 mm (Size 2) in Figure 18b. The surface chloride ion concentration is still 0.6, the transition layer thickness is still 75 μm, and the erosion time is still 88 d. The chloride ion distribution and average chloride ion concentration of specimen HPC50 with different aggregate size under the same depth are presented in Figure 19.

It can be seen from Figure 18 that when the aggregate size increases, and the proportion of aggregate remains unchanged, the aggregate number in concrete specimens will decrease, and the spacing between adjacent aggregates increases, which will be more conducive to the diffusion of chloride ions in the concrete.

Figure 19 also shows that when aggregate accounts is unchanged, only the particle size of aggregate is changed, the chloride ion concentration increases with the increase in aggregate size. With the increase in diffusion depth, the chloride ion concentration difference also increases gradually, the concentration difference is maximum when the diffusion depth is 15 mm. When the diffusion depth is greater than 15 mm, the concentration difference decreases, and the concentration tends to be equal due to at the depth 30 mm due to the limit of diffusion depth.

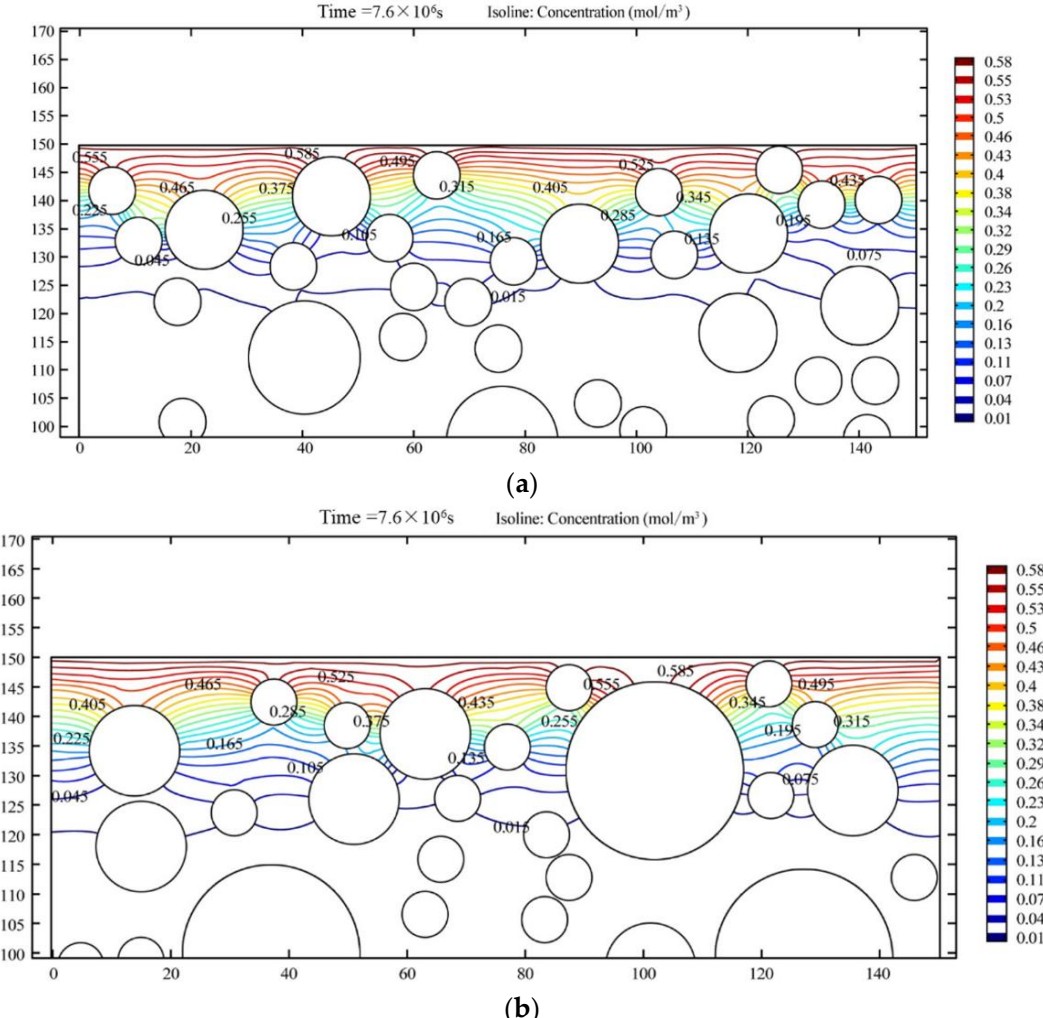

**Figure 18.** Chloride ion concentration distribution of specimen HPC50 with different sizes: (**a**) Size 1; (**b**) Size 2.

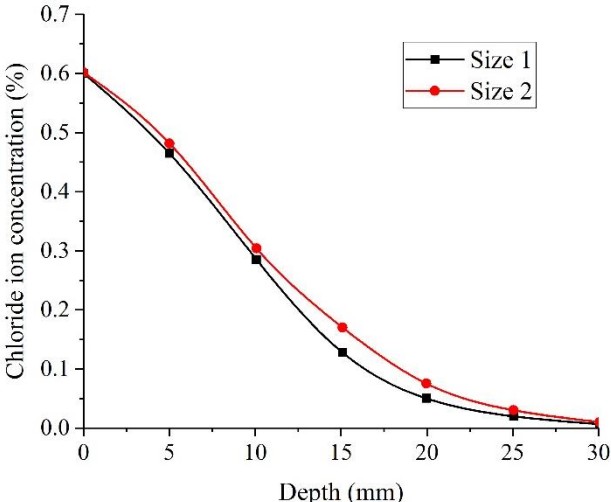

**Figure 19.** Chloride ion concentration of specimen HPC50 with different aggregate size.

## 8. Conclusions

In this paper, the chloride ion diffusion experiment for the HPC is carried out. The influence of W/B, binary (PC-FA and PC-SF), and binary (PC-FA-SF) combinations on the concrete compressive strength and chloride ion diffusion is investigated. The chloride ion diffusion coefficient is modified and the chloride ion diffusion model of the HPC is revised based on the Fick's second law considering the W/B, temperature, humidity, and time. A mesoscopic model is developed to investigate chloride diffusivity in concrete, the simulation results are compared with previous test data. The influence of aggregate shape, distribution, and size of HPC on the chloride ion diffusion is investigated. Some meaningful conclusions are obtained as follow:

(1) The influence of the W/B on the concrete strength for normal concrete is obviously more than the HPC. The contents of the SF or FA developing the compressive strength is limited. The concrete compressive strength of ternary combination specimens decreases with the increase in FA or SF when the content of the other mineral admixture SF or FA keeps unchanged.

(2) No matter normal concrete or the HPC, from depth 7.5 mm to 17.5 mm, the influence of W/B on the chloride ion concentration is most significant, under the same diffusion depth, the chloride concentration decreases with the increase in W/B. The ternary combination of PC-FA-SF is more efficient in prohibiting chlorides ingress inside the specimens than the binary combination of PC-FA or PC-SF.

(3) The mesoscopic simulation and the tested value of the chloride ion under the same depth is close, the average ratio of simulation value to tested value is 0.91. The aggregate shape and distribution have a negligible influence on chloride diffusivity in the HPC, but the chloride ion concentration increases with the increase in aggregate size.

**Author Contributions:** Conceptualization, H.T. and Y.Y.; methodology, H.T.; software, J.P. validation, H.T. and Y.Y.; formal analysis, H.T.; investigation, Y.Y.; resources, H.T.; data curation, H.T.; writing—original draft preparation, H.T.; writing—review and editing, Y.Y.; visualization, H.T.; supervision, J.P.; project administration, P.L.; funding acquisition, J.Z. All authors have read and agreed to the published version of the manuscript.

**Funding:** The work reported here was conducted with financial support from the National Natural Science Foundation of China (Grant Nos. 52108135 and 52078056), the National Key R&D Program of China (Grant No. 2021YFB2600900), National Science Foundation for Distinguished Young Scholars of Hunan Province (Grant No. 2022JJ10075), Natural Science Foundation of Hunan Province (Grant No. 2022JJ40024), the Science Fund for Creative Research Groups of Hunan Province (Grant No.

2020JJ1006), and the Scientific Research Outstanding Young Project of Education Department of Hunan Province (Grant No. 21B0721).

**Institutional Review Board Statement:** Not applicable.

**Informed Consent Statement:** Not applicable.

**Data Availability Statement:** All data, models, and code generated or used during the study appear in the submitted article.

**Conflicts of Interest:** The authors declare no conflict of interest.

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
