# Peer review of "Test and Mesoscopic Analysis of Chloride Ion Diffusion of High-Performance-Concrete with Fly Ash and Silica Fume"

_coatings, doi:10.3390/coatings12081095_

Round 1
Reviewer 1 Report
The paper is well presented and contains original results. However, authors are encouraged to improve their work based on the following comments:
1) Authors may explain deficiencies or shortcomings of other studies to make a bridge to introducing the novelty of their work.
2) The novelty of this work must be more explained.
3) For general readers, authors are encouraged to discuss other kinds of concrete structures such as: [(a) “Stability and dynamic analyses of SW-CNT reinforced concrete beam resting on elastic-foundation”].
4) More physical description should be added to the results and discussion section.
5) In conclusion, give only main findings of your research with an appropriate value.
Author Response
1) Authors may explain deficiencies or shortcomings of other studies to make a bridge to introducing the novelty of their work.
Answer: The deficiencies or shortcomings of other studies are added, please check the introduction.
2) The novelty of this work must be more explained.
Answer: The novelty is introduced on the last paragraph in the introduction.
3) For general readers, authors are encouraged to discuss other kinds of concrete structures such as: [(a) “Stability and dynamic analyses of SW-CNT reinforced concrete beam resting on elastic-foundation”].
Answer: This reference has been cited in the introduction.
4) More physical description should be added to the results and discussion section.
Answer: The influence of the W/B and the mixing method of FA and SF on the compressive strength and chloride concentration of HPC is discussed through experimental results, some meaningful conclusions are obtained.
5) In conclusion, give only main findings of your research with an appropriate value.
Answer: Some contents are deleted in the conclusion, the main finding in the experiment and me please check it
Reviewer 2 Report
The authors present a well-written paper related to an experimental campaign accompanied by numerical simulation in order to evaluate the chloride ion diffusion of High-Performance Concrete (HPC) with fly ash and silica fume. In this work the influence of W/B, binary (PC-FA and PC-SF) and binary (PC-FA-SF) combinations on the concrete compressive strength and chloride ion diffusion was investigated. However, some minor comments should be addressed:
11. The introduction section should be improved with more recent references existing in the literature.
22. The novelty and the major contributions of this work should be highlighted at the end of the introduction section.
33. Section 2.2 should be expanded with more details of the compressive strength test done.
44. Please compare the experimental results obtained with other presents in the literature.
55. Please explain why the authors used the ideal model-Fick’s second law instead the anomalous diffusion method or the Time-Fractional model
66. Please add bullets on page 15 after “The simulation steps are shown as follows:”
77. Please detailed with more “detail” the simulation method used (section 6.1)
Author Response
- The introduction section should be improved with more recent references existing in the literature.
Answer: The recent references are added in the introduction section, please check reference.
- The novelty and the major contributions of this work should be highlighted at the end of the introduction section.
Answer: The novelty and the major contributions of this work are highlighted in the introduction.
- Section 2.2 should be expanded with more details of the compressive strength test done.
Answer: The details of compressive strength test is added, please check the section 2.2.
- Please compare the experimental results obtained with other presents in the literature.
Answer: The compressive strength results of HPC in Sengul and Tasdemir 2009, Chindaprasirt et al. 2005 and Jerath and Hanson 2007s’ experiments are compared with this paper, please check sections 3.1 and 3.2.
- Please explain why the authors used the ideal model-Fick’s second law instead the anomalous diffusion method or the Time-Fractional model
Answer: Buenfeld et al (1998) considered that the progress of chloride transportation in the concrete is the unsteady diffusion process. For Fick’s second law, in unit time and unit volume, the mass of material flowing out is different from that flowing into the internal structure of concrete, and the concentration of any point in it varies with time and space, this progress is also the unsteady process. Many scholars (Mangat and Limbachiya 1999, Thomas and Bamforth 1999)concluded that the chloride ion diffusion process can be described by the Fick’s second law.
- Please add bullets on page 15 after “The simulation steps are shown as follows:”
Answer: The bullets are added, please check section 6.1.
- Please detailed with more “detail” the simulation method used (section 6.1)
Answer: More details about the meshing method of the simulation are added in section 6.1, please check it.